# Comparison of Different Pneumorrhaphy Methods after Partial Pulmonary Lobectomy in Dogs

**DOI:** 10.3390/ani13172732

**Published:** 2023-08-28

**Authors:** Paloma Helena Sanches da Silva, Carlos Eduardo Bastos Lopes, Larissa Bueno Stallmach, Lucas de Oliveira Ferreira, Pedro Antônio Bronhara Pimentel, Antonio Giuliano, Patrícia Maria Coletto Freitas, Rodrigo dos Santos Horta

**Affiliations:** 1Department of Veterinary Medicine and Surgery, Veterinary School, Universidade Federal de Minas Gerais, Belo Horizonte 31270-901, MG, Brazil; palomasanches.vet@gmail.com (P.H.S.d.S.); 1993carlos.eduardo@gmail.com (C.E.B.L.); laristall@gmail.com (L.B.S.); lucasoferreira.vet@gmail.com (L.d.O.F.); pedrobpimentel@gmail.com (P.A.B.P.); pcoletto@yahoo.com.br (P.M.C.F.); 2Department of Veterinary Clinical Sciences, Jockey Club College of Veterinary Medicine, City University of Hong Kong, Kowloon, Hong Kong, China

**Keywords:** cyanoacrylates, tissue glue, lungs, pneumostasis, surgical technique

## Abstract

**Simple Summary:**

Manual sutures, staples, and tissue adhesives are indicated for pulmonary surgery. However, despite its well-recognized use, no suture material has yet been pointed out as completely safe in preventing air leakage, a major complication of pulmonary surgeries both in humans and dogs. The aim of this study was to investigate the efficacy of the most common suture techniques and sealing devices after pulmonary lobectomy in dogs considering ventilatory pressures: physiological and supraphysiological. This study concludes that manual/mechanical sutures and synthetic glue sealants are efficient in preventing air leakage after partial lobectomy in dogs.

**Abstract:**

Pulmonary loborraphy can be performed using manual sutures and staples, although other methods, such as tissue adhesives, are also cited in the veterinary literature. Although the surgery is well tolerated in the canine species, failure in pulmonary aerostasis is still a reality since all the methods described so far eventually lead to air leakage after the use of the partial lobectomy technique in the lungs. Within this context, the aim of this research was to compare the effectiveness of different hermetic sealing methods after partial lobectomy of the right caudal lung lobe (RCLL) in dogs. 30 cadavers models were divided in 6 groups: G1—cobbler suture associated with simple continuous; G2—overlapping continuous suture associated with simple continuous suture; G3—Ford interlocking suture; G4—Stapling device; G5—Tissue glue (cyanoacrylate). After performing the sealing techniques, the lungs were submerged in water and inflated with oxygen at positive ventilatory pressures at physiological (up to 14.7 mmHg, which is equivalent to up to 20 cmH_2_O) and supraphysiological levels (above 14.7 mmHg) to evaluate the performance of the sealing methods. At physiological ventilatory pressure levels, there was no difference between groups. Sealing with surgical glue was superior to interlocking sutures and stapling devices at supraphysiological levels of ventilatory pressure.

## 1. Introduction

Pulmonary lobectomy is a surgical technique occasionally performed in dogs to resect pulmonary tumors, bullae, and damaged lung lobes [1,2,3]. This procedure consists of partial or total lobar resection, depending on the extension of the lesion in the parenchyma [1,2]. Although the surgical procedure using manual suture or stapling devices is usually safe, failures, including air leakage and consequent pneumothorax can occur [1,4,5]. Postoperative mild pneumothorax can resolve spontaneously or with the use of thoracic drains. However, when not treated, severe pneumothorax can result in severe complications and even death [1,3,4,6]. Various sealing techniques have been developed to increase the safety of the procedure. Specific tissue glue, as well as the use of different types of staplers, pre-tied loop ligatures, and vascular sealing devices, are now used in veterinary medicine [7,8,9]. However, it is not completely known which sealing method is the safest. Air leaking post-lung lobectomy is still a complication reported with most suture techniques and sealing devices [1,3,4,5,6,9]. The aim of this study was to investigate the airtight sealing property and efficacy of the most common suture techniques and sealing devices after pulmonary lobectomy against physiological and supraphysiological ventilatory pressures.

## 2. Materials and Methods

Thirty dogs’ cadavers were obtained after euthanasia following the diagnosis of leishmaniasis at the Zoonosis Control Center. Criteria for inclusion were: mixed breed dogs; weight between 5 and 10 kg; variable body condition score; macroscopically normal respiratory tract during necroscopy. The respiratory tract was removed en block, including the larynx, trachea, bronchi, and lungs. The experiment was carried out within 12 h after macroscopic collection of the respiratory block. An appropriately sized endotracheal tube was inserted into the trachea and secured with a nylon seal. The tracheal tube was connected to a calibrated manometer (Aneroid sphygmomanometer Premium; Accumed Produtos Médico Hospitalares, Duque de Caxias, RJ, Brazil) and oxygen gas for positive pressure. All lungs were initially pressurized slowly to 10 mmHg and then deflated to check for pre-existing atelectasis and/or previous air leakage before performing the experiment. The right caudal lung lobe (RCLL) was chosen, and the partial lobectomy was performed, followed by tissue sealing and aerostasis assessment. Different lung synthesis methods were organized into five experimental groups, each containing six experimental units: G1—Loborrhaphy with cobbler or Vachetta suture pattern, followed by simple continuous suture, with Polyglactin 910 suture thread (Coated Vicryl^®^; Ethicon, Johnson, and Johnson, São Paulo, SP, Brazil); G2—Loborrhaphy with overlapping continuous suture associated with simple continuous suture, with Polyglactin 910 suture thread; G3—Loborrhaphy with Ford interlocking suture pattern, with Polyglactin 910 suture thread; G4—Loborraphy with metal staples by TA linear stapler, two units 45 mm reload inserted in opposite directions into the lobe (TA^TM^ Stapler with DST Series^TM^; COVIDien, Minneapolis, MN, USA); G5—Loborraphy with 1 mL of synthetic adhesive containing n-butyl cyanoacrylate (NBCA) and methacryloisofolane (MS) (Glubran-2^®^; GEM srl, Viareggio, LU, Italy).

The width and height of the parenchyma in the middle third of the resected RCLL were measured with a caliper. For manual sutures, surgical threads of Polyglactin 910 of size 4–0 and a needle with a cylindrical body and the point of the needle taper point were used.

After performing the sealing techniques, the lungs were placed into a transparent plastic container and submerged in water. The lungs were then inflated, and all models were filmed to allow for accurate monitoring and recording of any potential air leak. A leakage was considered only when air bubbles were visualized at the level of the suture line or on the surface of the cut. Any occurrence of air leakage was recorded at physiological pressure (up to 14.8 mmHg or up to 20 cmH_2_O) or supraphysiological if above 14.8 mmHg. The value of air pressure at which any leakage occurred was also recorded.

For statistical analysis, the software GraphPadPrism v. 6.02 was used. Differences with a *p* value less than 0.05 (*p* < 0.05) were considered significant. The cadavers were randomized into five groups of six. The Kolmogorov–Smirnov test was used to assess the normality of the variables studied. The homogeneity of the groups was evaluated by comparing body weight, body score, height, and width of the RCLL. The comparison of the means between the groups of the parametric variable with normal distribution (body weight) was performed using ANOVA and Fisher’s ad hoc Tukey test. The comparison of the medians for the non-parametric variable (body score) and for those with non-normal distribution (height and width of the lobe) was performed using the Kruskall-Wallis ad hoc Dunns test. The pressure related to the air leaks also showed a non-normal distribution, and, therefore, the medians between the groups were compared using the same test. Furthermore, the groups were compared regarding the aerostatic variable, that is, whether or not there is air leakage under physiological ventilatory pressures considered up to 14.7 mmHg, using the Kurskall–Wallis ad hoc Dunns test. Fisher’s exact test was also used to evaluate the frequency of distribution in relation to leakage under physiological pressure, according to the sealing methods.

## 3. Results

All the five groups analyzed were homogeneous in terms of body weight (*p* = 0.1901) and lung lobe width (*p* = 0.0637). However, they differed in the median for the body condition score (*p* = 0.0202) and lung lobe height (*p* = 0.0347). Regarding the body condition score of the animals used in this study, there was a statistical difference only between the medians of groups G1 (cobbler suture followed by simple continuous suture) and G3 (Ford interlocking suture). Lung lobe height differed only between groups G4 (surgical staplers) and G5 (synthetic glue) (Figure 1).

The pressure in mmHg needed to cause lung lobe air leakage varied according to the type of sealing techniques (view in Appendix A).

Although some experimental units demonstrated a rupture in physiological pressures in groups G1, G3, and G4, there was no difference between the groups (*p* = 0.3263).

The pressure needed to cause air leakage in the lung lobe sutured with Ford interlocking suture (G3) and TA stapler (G4) was lower than the synthetic glue (G5), with *p* = 0.0164 (Table 1) (Figure 2). There was no statistically significant difference (*p* = 0.2873) in the sealing method’s efficacy against physiological pressure (up to 14.7 mmHg). However, while the G2 (overlapping continuous suture) and G5 (synthetic glue) did not have any air leakage at physiological air pressure, aerostasis failure was observed in 1/6 in s G1 (cobbler suture + continuous simple suture 2/6 in G3 (Ford interlocking suture), and 2/6 in G4 (stapling device).

## 4. Discussion

The physiological ventilatory pressure considered in this study was 14.8 mmHg (20 cmH_2_O). According to veterinary guidelines, pressure levels on lung tissue from 15 to 25 cmH_2_O can be considered physiological [10]. However, pressure values of 15 to 20 cmH_2_O are considered more accurate for small animals under normal respiratory mechanics conditions [11,12]. In one study, aerostasis was considered safe when air leakage occurred at ventilatory pressures greater than 20 cmH_2_O after a lung biopsy in anesthetized dogs [13]. In another experimental study evaluating different types of surgical staplers, air leakage at a ventilatory pressure equal to or less than 20 cmH_2_O was considered not safe and the sealing device inadequate [8]. Sealing methods probably should withstand a ventilatory pressure of the airways of up to 20 cmH_2_O to be considered safe. All the methods used for sealing that were investigated in this study were effective against established physiological ventilatory pressure.

The canine lung model used was macroscopically normal, but it was obtained from dogs with leishmaniasis. Most of the dogs were clinically unwell, which resulted in different body condition scores between the groups. The body condition score values [14] of G3 were lower (1–3) compared to G1 (3–7). However, there was no statistical difference in the aerostatic properties of these groups. Despite the difference between the median lung lobe height of G4 (Md = 12.12 cm) and G5 (Md = 19.47 cm), G5 showed superior sealing properties under supraphysiological pressures compared to the median obtained by G4, which proves that even with a larger surface area to be sealed, the synthetic glue was still able to withstand against high pressures ventilation, ensuring greater aerostatic safety than the staples device.

One of the manual sutures used and analyzed in this study was the cobbler suture [15] (Figure 3A), which ensured good sealing in almost all the lung lobes (Figure 3B). However, one developed air leakage at a pressure of 14 mmHg. The punctual air leak detected in the place where the suture was contained possibly corroborates the observation described by other authors when evaluating the double-layer mattress suture in sheep lungs after partial resection of one of the lobes [16]. The authors observed minimal air leakage along the parenchyma when the lungs were inflated to 40 mmHg and explained that this was due to the portion of the parenchyma along the needle incisions [16]. Thus, the small leak detected in the 1/6 model after the cobbler suture probably occurred when the needle passed through the tissue. However, there were no statistical differences between the models analyzed in the present study. The successful aerostasis observed in this group can be attributed to the increase in tissue contact surface caused by this suture pattern, which ensured sufficient sealing against physiological ventilatory pressures.

The overlapping continuous suture (Figure 4A) used in G2 has been described for lung sealing after partial lobectomy in dogs and cats [17,18]. This is a manual suture pattern defined as a pneumostatic suture once it results in additional compression of the parenchyma [18]. The results found in this study confirm the efficacy of this suture pattern in maintaining aerostasis against physiological ventilatory pressures (Figure 4B).

The third manual suture investigated (G3) was the Ford interlocking suture pattern (Figure 5A), characterized by a modified continuous suture and frequently used in soft tissue surgeries in order to withstand some degree of tissue tension [19]. Such sealing suture also demonstrated good aerostasis, although two of the models showed air leakage at physiological ventilatory pressures. It is worth mentioning that so far, there are no references in the veterinary literature regarding the use of the Ford interlocking suture pattern in loborraphy after partial lung lobectomy in dogs. This suture pattern is indicated for the correction of lacerations of the lung parenchyma in humans since it promotes good adhesion of its edges and, therefore, hemostasis [20]. This suture brings the margins closer, providing an effective seal. Despite the lack of studies with the Ford interlocking suture technique in dogs in vivo, this sealing technique could be a safe option after partial pulmonary lobectomy in dogs. Two out of six cadaver models in the Ford interlocking suture group failed to achieve aerostasis when the ventilatory pressure reached a value of 14 mmHg (Figure 5B). In both cases, along the suture line, there was no leakage, except for a single laceration point lateral to the suture line. When the ventilatory pressure was increased, the tissue tension in the lobe was probably higher in certain points, which may have caused a punctual laceration and consequent leakage of air at the site in a parenchyma which normally has a friable, fragile appearance [19,20,21].

In the group of lung lobes in which the mechanical suture was performed with metal staples using the TA linear surgical stapler, the results showed a good response in terms of aerostasis, with no difference compared to the other groups. However, two out of six cases failed due to air leakage laterally to the staple line at physiological ventilatory pressures of 10 mmHg and 14 mmHg. The effective aerostasis with mechanical stapling sealing devices observed in the present experimental study was also reported by other authors [1,22]. The safe use of staples has been previously demonstrated in nearly 40 dogs and cats undergoing pulmonary lobectomy for the treatment of different conditions [1]. In this study, only two patients developed pneumothorax in the postoperative period [1]. In an ex vivo study with lung biopsy in dogs, airway leakage occurred at a pressure of 20.5 mmHg. However, this was achieved using an EndoGIA 45 mm endoscope stapler [13]. In another study, also using a stapler during thoracoscopy, eight dogs were submitted to partial and total pulmonary lobectomies, and although the pulmonary inflation pressure was not mentioned, all animals were discharged without intraoperative complications after loborraphy with surgical staples [22]. In a clinical study, two main conventional methods of pulmonary sealing were compared after partial lobectomy in eight dogs. It was found that the mechanical suture with staplers was superior to manual suture with non-absorbable material [23]. The authors concluded that staples could be superior as they achieve an equal distribution along the parenchyma, while manual suture is more likely to depend on the surgeon’s technique, which may lead to variability in tissue sealing distribution [23]. Furthermore, the efficacy of surgical staples over manual sutures has been proven in humans undergoing pulmonary lobectomy [24]. The authors also believe that manual suturing depends on the surgeon’s experience and skills, and manual suturing could cause more air leakage caused by the passage of the needle throughout the pulmonary parenchyma [24]. However, in the present study, there was no statistical difference regarding the efficacy of aerostasis between manual and mechanical sutures. However, in the present study, aerostasis promoted using mechanical suturing was achieved by ensuring the insertion of staples for all the length of the sectioned parenchyma [5]. To avoid air leakage due to the failure of inserted staples, two blue cartridges with a total width of 45 mm each were used, containing a double row of staples (Figure 6A). The width × height dimension of each of these staples when opened was 4.0 × 3.5 mm. Both cartridges were inserted in opposite directions in the lobe in order to ensure that the lateral ends of the parenchyma to be stapled were all completely covered by staples, thus avoiding air leakage from uncovered areas. The presence of a wider cartridge, such as the 60 mm one, could have replaced the use of the two cartridges used in the study.

An interesting observation is that the leg length of the staple in mm did not match the thickness of the tissue to which it was inserted. According to the type of surgical stapler used in this study, the blue cartridge of the TA linear stapler has a leg length of 3.5 mm when open, with a staple height of 1.5 mm when closed [25]. The veterinary literature recommends that when surgical staples are used in the lung parenchyma, the selection of the height of the staples to be used depends upon the compressed thickness of the tissue [5]. The discrepancy of the height in vivo should be taken into consideration before deciding the type of staple to apply. However, in our study, lung lobes with different values of height or parenchyma thickness were used. In the stapling device group, the thinnest parenchyma was 10.89 mm, and the largest was 14.33 mm, which means that these values were higher than the standardized heights of the staples used in the study. Despite an incomplete cover of the staple over all the parenchyma to be sealed, air leakage did not occur since the six lobes of G4 did not show differences at physiological pressures. Probably, the reduced length of the staple, when closed, should consider only the diameter of the bronchioles and not the thickness of the tissue to ensure aerostasis [5]. However, the use of staples in pulmonary loborrhaphy in dogs, although considered safe, does not rule out completely the risk of air leakage during intraoperative positive pulmonary ventilation, as observed in two out of six models in the staples group. Air leakage has been documented at the staples site under physiological ventilatory pressures in other studies with dogs’ lungs [8]. In a study evaluating three different models of surgical staplers used in the pulmonary loborrhaphy of canine cadavers, the rupture pressure at the stapling line seldom occurred at values equal to or less than 20 cmH_2_O (14.8 mmHg) [8]. Air leakage along the stapling line was reported by other authors when using an EndoGia 45 mm endoscopic stapler [13]. When performing lung lobe biopsy in vivo in dogs and comparing Endo Gia 45 mm with vascular sealing devices and sutures, an air leak was found at the stapler site, possibly caused by staple trauma to the parenchyma. The author concluded that the stapler device was considered safe because the median pressure values of air leakage were not less than 20 cmH_2_O [13]. At high positive pressure, the inserted staples can lacerate the lung tissue, resulting in wider orifices and consequent air leakage [26], even at pressures around 20 to 25 cmH_2_O. This is the likely explanation for what may have caused air leakage in the two models in the staples group (Figure 6B). Although the superiority of mechanical suturing has been reported, an additional manual suture over the staple line used for partial pulmonary loborrhaphy may be necessary to control parenchymal leakage and bleeding [23]. This suture reinforcement in the stapling line is also recommended by other authors and can be performed with additional stapling or with manual suturing over the area of air leakage [5,23,25]. Complementing the staple line with other surgical sutures or devices could guarantee better resistance at supraphysiological levels of ventilatory pressure [26,27,28]. However, this was not evaluated in our study.

The effective sealing property provided by the Glubran-2^®^ surgical sealant (G5) did not result in aerostatic failure in any of the six lungs when challenged by physiological ventilatory pressure. Air leakage occurred only at very high ventilatory pressure, with a minimum of 20 mmHg and a maximum of 80 mmHg (Figure 7A,B).

Similar results were achieved in an experimental study using rabbit lungs and sealant that contains NBCA, one of the monomers present in Glubran-2^®^ [29]. In such study, with rabbits the authors demonstrated that of the 12 lung lobes resection, air leakage developed only at supraphysiological ventilatory pressures, ranging from 18.38 to 40.45 mmHg (25 to 55 cmH_2_O) [29]. Similarly, in our study, sealing failure occurred only at pressure above 20 mmHg, demonstrating greater safety after pulmonary loborrhaphy in dogs. In the present study, two of the six lobes in the glue group had aerostasis failure at 60 and 80 mmHg, 5.4 times higher than the physiological pressure and above the maximum pressure recorded in the study with rabbit lungs [29]. Another similar result was reported in an experimental study that used the same type of sealant in 20 sheep lungs against supraphysiological pressures of 40, 60, and 80 mmHg [16]. The authors noted that most of the lobes did not present air leakage in the region sealed using the glue, except for a smaller percentage of 20% of the lobes that resulted in a minimal degree of air leakage.

The Glubran-2^®^ sealant is one of the commercial names of NBCA, an important synthetic biodegradable surgical glue, which is modified using the addition of MS monomer [30,31,32]. It has been widely used in medicine, such as in urological, gynecological, digestive, neurological, cardiovascular, and thoracic surgeries because it demonstrates adhesive, hemostatic, and aerostatic properties when applied to different tissues [31,33,34]. The safety and efficacy of this synthetic glue have been reported in cats, swine, rabbits, sheep, and humans [9,16,30,33,34,35,36]. However, so far, no reports have been described or published using the cyanoacrylate co-monomer Glubran-2^®^ in surgeries involving lobectomy in dog lungs. In a study with six healthy cats after performing partial pulmonary lobectomy, NBCA has effective hemostatic and aerostatic properties [9]. The results were satisfactory, although the ventilatory pressure used to assess aerostasis was not reported, nor was the extent of resected parenchyma. However, due to the absence of clinically significant post-surgical complications and the histopathological evaluation of the patches containing the glue three weeks after surgery, it was concluded that this type of cyanoacrylate would be a safe alternative method for pulmonary loborrhaphy in cats [9]. From radiographic examinations, mild asymptomatic pneumothorax was observed only in 4 patients soon after surgery and in 1 patient two days post-surgery [9]. The sealing property of the glue is due to the formation of a thin elastic film of high strength that molds itself over the tissue and makes the tissue region firm within 60 to 90 s. This is the result of the glue’s exothermic polymerization after interacting with the tissue surface, forming strong bonds with tissue proteins and ensuring strong sealing to the lung tissue at the place where it is applied [30,32,36]. In the experimental study with rabbit lungs, local and systemic effects were investigated after the use of the synthetic cyanoacrylate sealant [36]. It was macroscopically observed that the portion of the lung parenchyma that was glued had an altered texture and hardened consistency for almost a month after being applied [36]. This highlights the efficacy of the sealing property of this product for pulmonary aerostasis. In the current study, although no difference was observed between the groups regarding rupture under physiological pressure, the sealing promoted using the synthetic glue containing NBCA was able to ensure efficient aerostasis at supraphysiological ventilation pressures. The median pressure of rupture in G5 (Glubran-2^®^) was superior to that observed in G3 (Ford interlocking suture) and G4 (staples). In addition, if the surgical time variable had been evaluated in the different groups studied, sealing with surgical glue would probably stand out in reducing surgical time due to its application method and chemical property, reaching maximum mechanical force on the tissue within 1 to 2 min [30,31,32].

In the present study, the canine lung lobes of the three groups that received the different manual sutures made with the same type of surgical material were not able to achieve aerostasis at pressure levels of 40 mmHg. Similarly, as demonstrated in the study carried out with sheep lungs after partial lobectomy, all lobes that received manual sutures failed aerostasis after applying this pressure [16]. However, among the manual sutures analyzed in the current study, only the group that received the Ford interlocking suture showed a significantly lower difference compared to the group that received the synthetic glue. The air leak observed in the manual suture line proved to come from the place where the needle passed into the parenchyma [16]. Similar to our study, the authors concluded that the aerostasis promoted using the sealing performed by the manual suture was inferior to the sealing performed by the synthetic glue in all 20 sheep lungs evaluated at high ventilatory pressure [16].

In our study, synthetic cyanoacrylate adhesive was significantly superior to the manual Ford interlocking suture and to mechanical suturing by staples. The excellent seal maintained by the synthetic glue can be explained by the low viscosity of the sealant when in the liquid phase, filling the small gaps and creating better sealing between the cutting surfaces of the lung than other sealing methods [32]. The glue also does not create physical injury to the lung parenchyma, as the tissue trauma caused by the needle and staples [13,16,24].

The main limitation of this study is the experimental model. The lungs were evaluated outside the thoracic cavity, a condition that prevents adequate lung distension and retraction during the respiratory cycle [37]. The results presented and the findings observed in each experimental group of this study refer to an ex vivo experimental condition, a situation that is different from the respiratory dynamics normally observed in live animals in which lungs are continuously expanded by air under the influence of different ventilatory pressures. Under laboratory conditions, accurate assessment of the sealing methods against physiological variations of ventilatory pressures is not completely possible. Furthermore, the lack of active bleeding at the site, due to the experimental model being ex vivo, tends to facilitate the application and adhesion of the surgical glue. However, the adhesive used in the experiment allows adhesion even in a humid location. One of its properties is that, in addition to its aerostatic property, it also has a hemostatic property (variable not evaluated in this ex vivo study), not being altered in a liquid medium (blood or other organic fluid), according to manufacturers and studies with this type of combination of cyanoacrylate. Another point to remember is that on post-mortem examination, the lungs appeared macroscopically normal, and the experiment was carried out within a few hours after death. However, this is not completely comparable to a lung in a physiological condition. Therefore, the safety and efficacy data of aerostasis in all the methods evaluated in this experimental model should not be completely and safely extrapolated for dogs in vivo. Another limitation regards the collection of lungs from patients diagnosed with leishmaniasis. It is possible that microscopic lung lesions could be present at the time of the study without being macroscopically visible. *Leishmania* sp. is known to cause histological changes, including mild to severe peribronchial inflammatory infiltration, fibroblast proliferation in lung tissues, edema, and congestion in the alveolar wall, but histological examination of the lungs in this study was not performed [38]. Microscopic changes, whether caused by *Leishmania* sp. or some subclinical respiratory disease, could interfere with lung compliance and adequate sealing, resulting in rupture at lower pressures, and could justify some results of this study [6,13,38].

## 5. Conclusions

The result of this study proved the safety and efficacy of all investigated sealing methods after partial lobectomy in the middle third of the right caudal lung lobe of dogs. There was no difference in physiological ventilatory pressures. However, greater efficacy was achieved by the surgical tissue glue Glubran-2^®^ at supraphysiological ventilatory pressures (greater than 14.7 mmHg). Glubran-2^®^ was superior to the Ford interlocking suture and the TA 45 mm linear surgical stapler, but no difference was found between Glubran-2 and cobbler suture associated with simple continuous and overlapping continuous with simple continuous suture.

## Figures and Tables

**Figure 1 animals-13-02732-f001:**
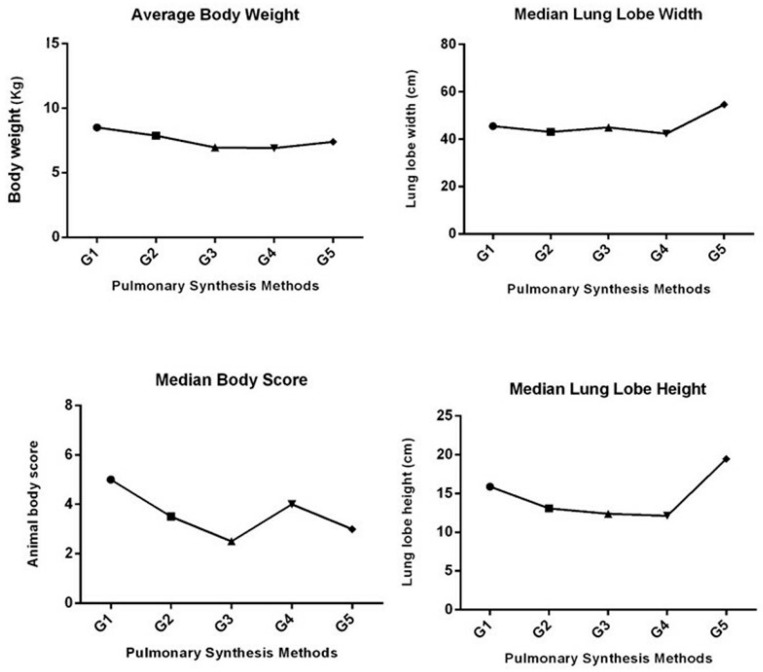
Mean and median values of the variables analyzed in the different groups (G1 = cobbler + simple continuous; G2 = overlapping continuous + simple continuous; G3 = Ford interlocking suture; G4 = staples; G5 = synthetic adhesive). Mean for the variable body weight (Kg), *p* = 0.1901. Medians for the variable lung lobe width (cm), *p* = 0.0637. Medians for variable body score of the animals, with the statistical difference between G1 and G3, *p* = 0.0202. Medians for the variable lung lobe height (cm), with statistical differences between G5 and G4, *p* = 0.0347.

**Figure 2 animals-13-02732-f002:**
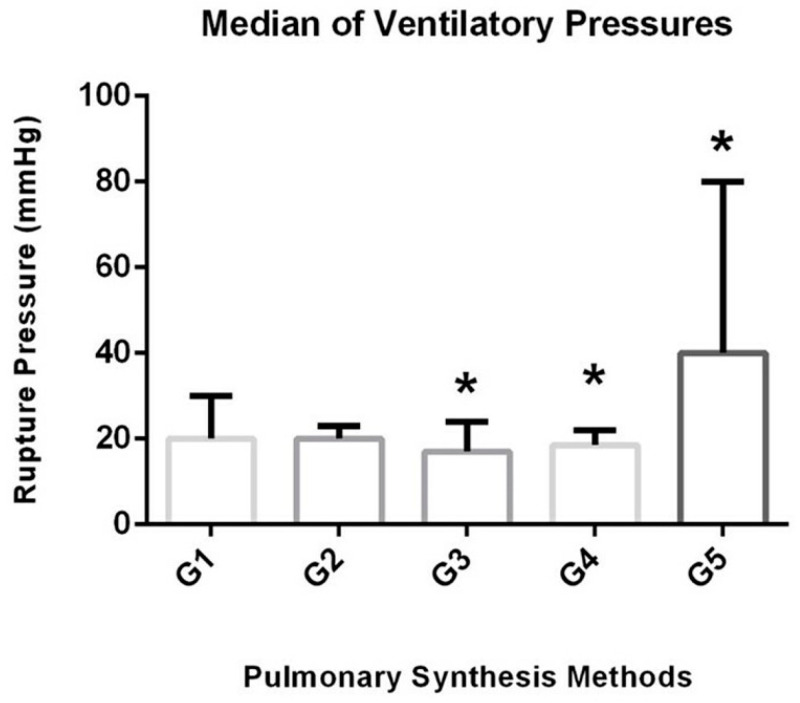
Median burst pressures (mmHg) referring to the types of synthesis in which there was loss of pulmonary aerostasis, *p* = 0.0164. G1 = cobbler + simple continuous; G2 = overlapping continuous + simple continuous; G3 = Ford interlocking suture; G4 = staples; G5 = synthetic glue. * indicates significance level at 5%. The groups differ from each other at a 5% significance level.

**Figure 3 animals-13-02732-f003:**
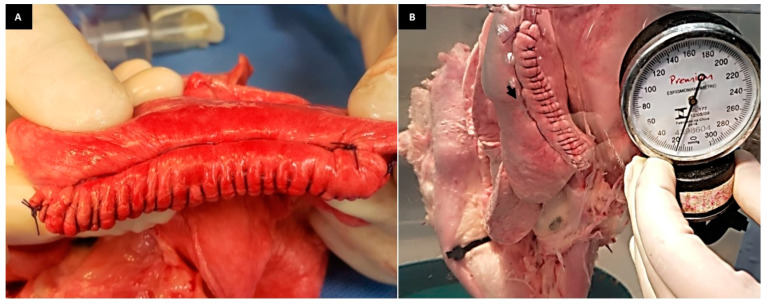
(**A**): Photographic image of the cobbler and simple continuous suture in the lung lobe. (**B**): Photographic image of the RCLL (G1) showing air leakage (arrow) laterally to the suture line at supraphysiological pressure at 18 mmHg.

**Figure 4 animals-13-02732-f004:**
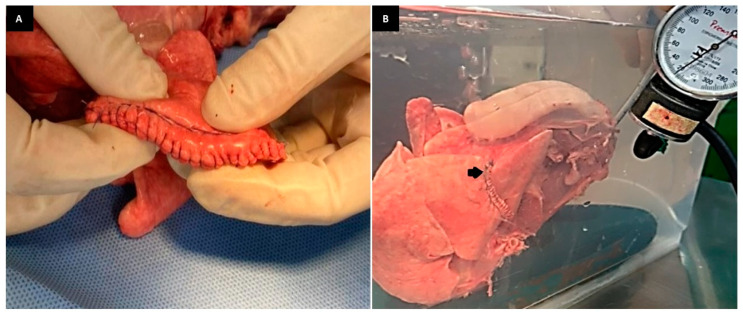
(**A**): Photographic image of the overlapping continuous and simple continuous suture in the lung lobe. (**B**): Photographic image of the RCLL (G2) showing air leakage (arrow) at supraphysiological pressure at 20 mmHg.

**Figure 5 animals-13-02732-f005:**
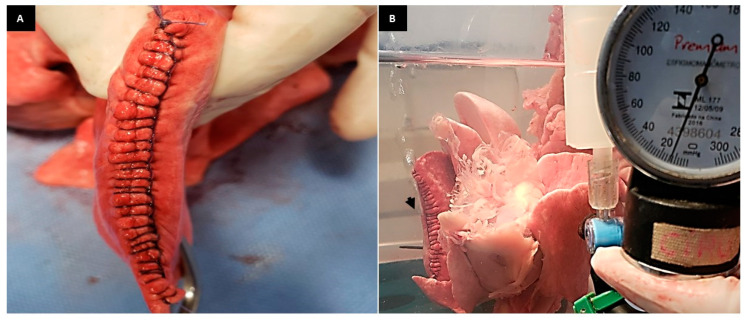
(**A**): Photographic image of the Ford interlocking suture in the lung lobe. (**B**): Photographic image of the RCLL (G3) showing air leakage (arrow) at supraphysiological pressure at 14 mmHg.

**Figure 6 animals-13-02732-f006:**
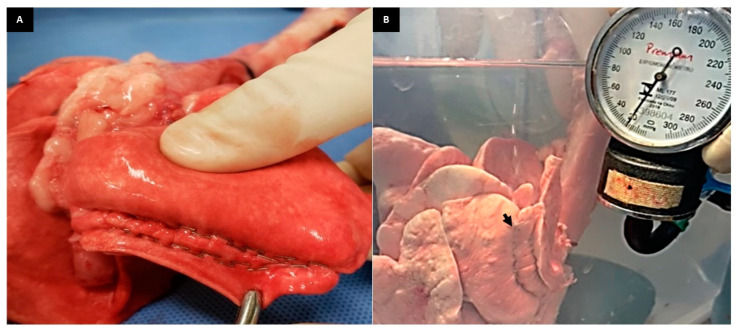
(**A**): Photographic image of the double row of staples in lung lobe lung. (**B**): Photographic image of the RCLL (G4) showing air leakage (arrow) at supraphysiological pressure at 19 mmHg.

**Figure 7 animals-13-02732-f007:**
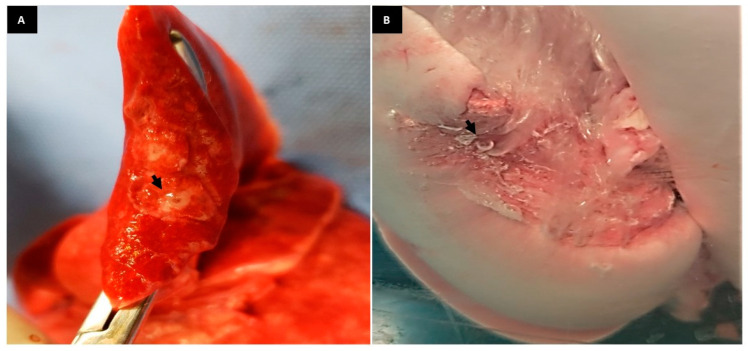
(**A**): Photographic image of the Glubran-2 adhesive in the lung lobe. Note the sealant in bronchioles (arrow). (**B**): Photographic image of the RCLL (G5) showing air leakage (arrow) at supraphysiological pressure at 80 mmHg in a resected area during the aerostasis test.

**Table 1 animals-13-02732-t001:** Values of the mean, median, standard deviation, and 95% confidence interval for the values of the groups analyzed regarding the variable extravasation pressure.

Groups	Mean	Median	Standard Deviation	Confidence Interval 95%	
				Low	Higher
G1 (Cobbler suture + simple continuous suture)	20.33	20.00	5.279	14.79	25.87
G2 (Overlapping continuous suture + simple continuous suture)	19.50	20.00	2.345	17.04	21.96
G3 (Ford interlocking suture)	17.83	17.00 ^b^	4.167	13.46	22.21
G4 (Staples)	17.17	18.50 ^c^	4.401	12.55	21.78
G5 (Synthetic glue)	46.67	40.00 ^a^	20.66	24.99	68.34

There was a difference between G5 in relation to G3 and G4, with *p* = 0.0164. Medians followed by distinct letters (^a, b, c^) differ from each other at a 5% significance level.

## Data Availability

All data generated or analyzed during this study are included in this published article and its Appendix A.

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
