# Peer review of "Comparison of Different Pneumorrhaphy Methods after Partial Pulmonary Lobectomy in Dogs"

_animals, 2023, doi:10.3390/ani13172732_

Round 1

Reviewer 1 Report

Dear Authors,

I appreciated reading this novel research, I think that this can be the first step to a further in vivo lobectomy comparison.

I think that also the surgical time between the different groups should be highlighted, thinking that gluing was faster than other groups.

I have minor comments:

Please revise the bibliography following Journal guidelines.

Please check spacing through the paper:

es: "However" line 44

Line 81 and 88: ad-hocTukey and Dunns

Remove Line 10: Efficient

Add references: Line 37-38 "consequent pneumothorax"

RCLL: write in extenso the first time over the abstract

Table 1: add manufacturer data in the legend (Covidien, etc) 

Table 2: check the title raw, column G3

Reviewer 2 Report

The manuscript “Comparison of Different Synthesis Methods After Partial Pulmonary Lobectomy in Dogs” by Paloma Silva et al, seems to be an interesting manuscript on the efficacy of some common suture techniques and sealing devices after pulmonary lobectomy in dogs. To start with, the term “synthesis” used in the title is improperly used. Synthesis, a Greek word, has a lot of meanings; one of them being “the composition or combination of parts in order to form a unity” but certainly not the meaning the authors seem to imply, i.e. sealing or closure of a lung after lobectomy or, as currently said, pneumonorrhaphy. Therefore, the manuscript’s title must change. Furthermore, I have some rather major concerns about the study design and methodology used. First of all, did the authors perform a power analysis before deciding on the number of lungs to be used in the study? The fact that not many statistically significant differences were found might be due to their choice to  divide  a rather small sample of lungs into many groups. Another concern of mine is that the equipment used might not be appropriate for the accurate measurement of air pressure. Perhaps, a digital manometer would be much more accurate during measurements. Furthermore, the authors should mention that their findings concern ex vitro experimentation with one-time lung expansion and may not be consistent with the response of a lung that is continuously expanded by air in a living dog. Finally, I do not believe that you can have as a citation in your paper the same paper as a pre-print (reference 38). A number of other corrections can be found in the manuscript as yellow underlining, along with some more comments.

Some English language editing is required

Reviewer 3 Report

Dear authors,

The authors reported on various suturing techniques, staplers, and tissue adhesives regarding their effectiveness in preventing air leakage after partial lung lobectomy under physiological and supraphysiological pressure conditions. The findings of this study raise some questions about the interpretation of the results, as there are discrepancies from the clinical case reports mentioned in the authors' discussion. For instance, in most clinical cases, TA staples tend to recover well without adverse effects such as air leaks. However, since there is no direct comparative report of these methods, this study could serve as a valuable reference. For better reader comprehension, I kindly request a few modifications.

General Concept Comments:

Q1. I would appreciate it if you could mention the reasons for selecting Vicryl (multifilament) sutures. While many reference books recommend the use of absorbable sutures for partial lung lobectomy, I am curious about the specific rationale behind choosing multifilament sutures for the experiment. Generally, multifilament sutures exhibit excellent frictional force but are often avoided in tissues with infection concerns. Moreover, they tend to induce more tissue damage compared to monofilament sutures. These properties of sutures could potentially contribute to the occurrence of air leaks at lower pressures through the needle holes, as seen in your experimental results.

Minor point:

Q2. Please add your references after the following sentences:

[p9 L299-300]; [p9 L312-313]; [p10 L332-333]; [p10 L333-334]

Q3. [Table 2] It is deemed unnecessary to present all the low data in the main body of the paper. It would be advisable to provide them as supplementary data.

Q4. [Table 3] Could you kindly annotate the meaning of a, b, and c in the table? Furthermore, I believe it would be more beneficial to provide the detailed contents of the 'Group' in the form of annotations rather than directly within the table, in order to enhance reader-friendliness.

Q5. Generally, figures in a paper are employed to illustrate experimental methods or results. So, I believe it would be less desirable for figures 3-7 to be initially mentioned in the discussion section. It would be more appropriate for them to be presented in the materials and methods section or the results section.

Additionally, please provide legends for Figure 3.

Q6. "Would you kindly review the document for any typographical errors?"

* Examples of some misspelled words…

      [p1 L7]  university university?

      [Table 1] borraphy loborraphy

      [p5 L145]  wasmacroscopically was macroscopically, butit but it

Round 2

Reviewer 2 Report

Even though the authors did not include information about a power analysis that should have been conducted before determining the number of lungs to use in the study, they still improved the manuscript. 

It is advisable for the authors to perform some English editing, such as using dots instead of commas for numbers with decimals.
